# Carious lesions in permanent dentitions are reduced in remote Indigenous Australian children taking part in a non-randomised preventive trial

Ratilal Lalloo[1]*, Santosh K. Tadakamadla[2], Jeroen Kroon[2], Lisa M. Jamieson[3], Robert S. Ware[4], Newell W. Johnson[2,4,5]

1 School of Dentistry, The University of Queensland, Herston, Queensland, Australia, 2 School of Dentistry & Oral Health, Griffith University, Gold Coast, Queensland, Australia, 3 Adelaide Dental School, University of Adelaide, Adelaide, Australia, 4 Menzies Health Institute Queensland, Griffith University, Gold Coast, Queensland, Australia, 5 Faculty of Dentistry, Oral and Craniofacial Sciences, King's College London, London, United Kingdom

* r.lalloo@uq.edu.au

**Data Availability Statement:** Because of ethical considerations in respect of Indigenous communities, access to raw data has to be

## Abstract

We tested the effect of an annual caries preventive intervention, delivered by a fly-in/fly-out oral health professional team, with Indigenous children residing in a remote Australian community. Around 600 Indigenous children aged 5 to 17 years were invited to participate at baseline, of who 408 had caregiver consent. One hundred and ninety-six consented to the epidemiological examination and intervention (Intervention group) and 212 consented to the epidemiological examination only (Comparison group). The intervention, which occurred annually, comprised placement of fissure sealants on suitable teeth, and application of povidone-iodine and fluoride varnish to the whole dentition, following completion of any necessary restorative dental treatment. Standard diet and oral hygiene advice were provided. Caries increment (number of tooth surfaces with new dental caries) in both deciduous and permanent dentitions was measured at the 2-year follow-up. Comparison group children had significantly higher number of new surfaces with advanced caries in the permanent dentition than the Intervention group (IRR = 1.61; 95% CI: 1.02–2.54; p = 0.04); with a preventive fraction of 43%. The effect of intervention remained significant with children in the Comparison group developing significantly more advanced caries lesions in the permanent dentition than the Intervention group children in the adjusted multivariable analysis (IRR = 2.21; 95% CI: 1.03–4.71). Indigenous children exposed to the intervention had less increment in advanced dental caries in the permanent dentition than those not exposed to the intervention.

## Introduction

Globally, oral diseases represent one of the most prevalent chronic conditions, this burden largely unchanged for 25 years [1]. Untreated oral disease increased by a billion people: 2.5 in

approved by the University Ethics Committee, who will consult the community itself. Access may be requested via the Lead Chief Investigator E/Prof Newell Johnson at n.johnson@griffith.edu.au or directly from the Griffith University Human Research Ethics Committee (Ref: DOH/05/15/HREC) at research-ethics@griffith.edu.au, with a copy to E/Prof Johnson. All authors have full access to the all data (including statistical reports and tables) and gave final approval and agree to be accountable for all aspects of the work.

**Funding:** Funded by Australian National Health and Medical Research Council [NHMRC] Project Grant APP1081320 "Effectiveness, cost-effectiveness and cost-benefit of a single annual professional intervention for the prevention of childhood dental caries in a remote rural Indigenous community." Funding was to all authors.

**Competing interests:** The authors have no competing interests.

1990 to 3.5 billion in 2015 [1]. In 2015, the global cost due to dental conditions was estimated at over half-a-trillion dollars [2]. The dominant downstream approach is making little impression on the burden of oral conditions at the population level. Upstream approaches of addressing fundamental social determinants and drivers of disease have enjoyed little traction.

The 2012–14 National Child Oral Health Survey, Australia, showed that among Aboriginal and Torres Strait Islander children (hereinafter respectfully referred to as Indigenous children) in remote communities, the mean number of decayed, missing or filled surfaces (dmfs) in the deciduous dentition was 7.3, of which 5 were decayed. Almost two-thirds (64%) of remote-living Indigenous children experienced dental caries [3]. In the permanent dentition, older remote Indigenous children had mean DMFS of 2.5, with 1.7 surfaces decayed; 59% with caries experience [3]. This burden is replicated in adult Australian Indigenous people; and across Indigenous communities globally [4].

We previously reported the effectiveness of water fluoridation on dental outcomes in a remote Indigenous community in Far North Queensland [5]. Water supplies were fluoridated across the community in 2005. Prior to this, all school-going children were surveyed, demonstrating caries experience of 6- and 12-year-olds more than twice that of the Queensland average, and more than four-times greater than Australian children overall [6]. A follow-up survey in this community, in late 2012 by our research team, in which over 70% of known resident schoolchildren (n = 339) were examined, showed the dental caries status had improved significantly since the pre-water fluoridation survey [5]. Unfortunately water fluoridation ceased mid-2011.

Because caries remained a major problem, and would likely worsen without water fluoridation, our team successfully engaged with the Indigenous community and gained funding to test the effectiveness and cost-benefit of a caries preventive intervention [7]. This involved selective placement of fissure sealants, swabbing the dentition with povidone-iodine and application of fluoride varnish (termed a 'Big Bang' intervention). These individual preventive interventions have been found effective in reducing caries incidence [8, 9], but the combined effect is uncertain. A comparison of fluoride varnish and fissures sealants in a randomized clinical trial showed no significant difference in the number of newly decayed teeth over 3 years [10]. A program comparing primary and secondary prevention (glass ionomer sealants and interim therapeutic restorations) and one primary prevention only (glass ionomer sealants) showed no significant difference in reducing total caries experience but new caries experience was slower in the former intervention [11]. While fissure sealants are retained for a reasonable time and usually do not require re-application, fluoride varnish and povidone-iodine are most effective when applied 2–3 times a year [8]. In resource-constrained communities this is unsustainable. The overall purpose of the study was to determine whether professional interventions known to be effective individually, but deliberately applied together for maximum effect, compared to care as usual, were associated with a decrease in caries incidence and were cost-effective. The aim in the analysis presented here was to assess the effectiveness of an annual intervention on caries increment at the 2-year follow-up. We hypothesised that children who received the intervention would experience significantly fewer new carious lesions compared to those who did not.

## Methods

Following extensive consultation with the community, it became apparent that a non-randomised trial would be the most culturally appropriate approach. Ethics approval was granted by Griffith University (GU Ref: DOH/05/15/HREC); Far North Queensland (FNQ HREC/15QCH/39-970); Department of Education and Training (Queensland Government) to

approach participants at schools; and Torres and Cape Hospital and Health Service for Site Specific Approval. The research is registered with Australian New Zealand Clinical Trials Registry (ACTRN12615000693527: 3rd July 2015). This study is reported following the Transparent Reporting of Evaluations with Non-randomised Designs (TREND) guideline [12].

A detailed protocol is published [7]. All (approximately 600) school-going children (aged 5–17 years) in the community were invited to participate, by virtue of a detailed information sheet with detachable consent form, to be signed by a parent or guardian. At baseline, in the year 2015, approximately two-thirds of the children (N = 408) consented to a head and neck, and oral epidemiological examination. For those in need of dental treatment, separate signed consents were obtained according to Queensland Health standard procedures. The preventive intervention, delivered by a fly-in clinical team, comprising placement of fissure sealants, application of povidone-iodine and fluoride varnish, followed completion of individual treatment plans. Incentive to participate was provision of a well-organised and easily accessible treatment program at a mobile unit located at one primary school and at the dental clinic in the hospital, which is within walking distance of the community high school. Transport was provided for children who attended the second primary school a few kilometres away. In 2015 a clinical team comprising a dentist, oral health therapists and dental assistants lived in the community for three months. A smaller clinical team visited at first follow-up. Of the 408 children, those who consented to dental treatment (if required) and to the preventive intervention comprised the Intervention group, while those who did not formed the Comparison group.

## Outcome variable

Dental caries status of children was assessed by four trained and calibrated examiners, using the International Caries Detection and Assessment system (ICDAS-II) [13]. Children were examined in specially set-up classrooms, using mobile, reclinable chairs with fixed- and head-LED lights. Disposable mouth mirrors and blunt probes were used, as was gauze control moisture. All examiners completed ICDAS-II online training. For inter-examiner agreement, 5% of children were re-examined by another examiner, and discrepancies discussed with the child present. Overall kappa was 0.84, indicating high agreement. The epidemiological team visited the community for ~3-weeks each year of the study—usually September/October after the Wet Season.

Caries increment at the 2-year follow-up was the primary outcome. Any surface caries free at baseline examination (2015) but observed to have caries at the follow-up examination in 2017 was considered new caries/incident caries, and used for calculating caries increment (number of surfaces with new caries in each participant). Surfaces caries free at baseline but found to be extracted due to caries in subsequent examination, were also included when calculating caries increment, with all surfaces of an extracted tooth counted as missing. For analysis, caries increment was quantified for incipient (early stages of dental decay, identified as a white-spot lesion; ICDAS 1 and 2 scores) and moderate-advanced caries (ICDAS 3–6), separately and together (total caries increment) in the deciduous and permanent dentitions.

## Statistical analysis

Summary statistics are presented for continuous data as either mean (standard deviation) or median (interquartile range) as appropriate, and for categorical data as frequency (percentage). The association between group (Intervention/Comparison) and baseline characteristics was investigated using either the Mann-Whitney U test or the Chi-square test. For the outcomes—incipient caries increment, advanced caries increment and total caries increment—the preventive fraction was calculated as the percentage of the ratio of the difference between the mean

increment in the Comparison and Intervention groups to the mean increment in the Comparison group. The preventive fraction is an estimate of the percentage of carious lesions able to be prevented if children belonged to the Intervention rather than the Comparison group.

The association between group and caries increment was investigated using negative binomial regression models. In the initial model, group was included as the main effect and baseline caries experience as the covariable. Multivariable analyses with full adjustment for all covariables were conducted for the caries outcomes that demonstrated significant associations with group in the initial model. Covariables included in the adjusted models were age, sex, baseline caries experience, soft drink, lolly, syrup/jam, sugar consumption, salivary mutans streptococci and salivary LB levels. The negative binomial model was chosen due to the overdispersion of the outcome data, and the consequent inappropriateness of a Poisson model for this count data. Effect estimates are presented as Incidence Rate Ratios (IRR) with 95% confidence intervals (CIs). Due to the study design, the primary analysis was undertaken on a per-protocol basis.

To test the sensitivity of our results, we also completed an intention to treat (ITT) analysis. For the ITT analysis missing data was imputed using a fully condition-specification markov chain Monte Carlo method [14]. Data were assumed to be missing at random, ten iterations were used, and five imputed datasets were obtained. Variables with complete data (group and age) were also included in the model. Missing interval data was imputed using linear regression and missing binary data with logistic regression. Data were constrained by their minimum and maximum values. Both the initial (that is, with group and baseline caries experience included as main effects) and the adjusted model were calculated. A p-value of <0.05 was considered statistically significant. Data analysis was undertaken using SPSS (IBM Corp. 2016. v24.0. Armonk, NY).

## Results

Of the 408 children who consented to participate, at 1-year follow-up, 65% of children were retained (67% Intervention, 64% Comparison). At 2-year follow-up, 51% were retained (60% Intervention, 43% Comparison) (Fig 1).

At baseline, children in the Intervention and Comparison groups were similar in terms of sex, age, oral hygiene, salivary physiology and microbiology (Table 1). Children in the Comparison group were more likely to report adding sugar to cereal and drinks (P = 0.01). Children in the Comparison group were more likely to have incipient caries in the permanent dentition (P<0.001) and advanced caries in the deciduous dentition (P = 0.03). However, importantly the burden of advanced caries in the permanent dentition were similar in the groups.

At the 2-year follow-up, which is the focus of this analysis, children in the Intervention group had fewer new tooth surfaces with caries lesions, although not all comparisons were significantly different (Table 2). Comparison group children had significantly higher new surfaces with advanced caries in the permanent dentition than the intervention group (IRR = 1.61; 95% CI: 1.02–2.54; p = 0.04); with a preventive fraction of 43%.

Table 3 presents the multivariable analyses with the increment of advanced caries in the permanent dentition at follow-up as the outcome, controlling for potentially confounding background characteristics. The effect of group allocation was significant (IRR = 2.21; 95% CI: 1.03–4.71; p = 0.04). Age, salivary levels of MS and LB were the other variables that were significantly associated with caries increment. Caries increment increased with increase in age while those with lower levels of salivary MS and LB levels had fewer new carious surfaces (Table 3).

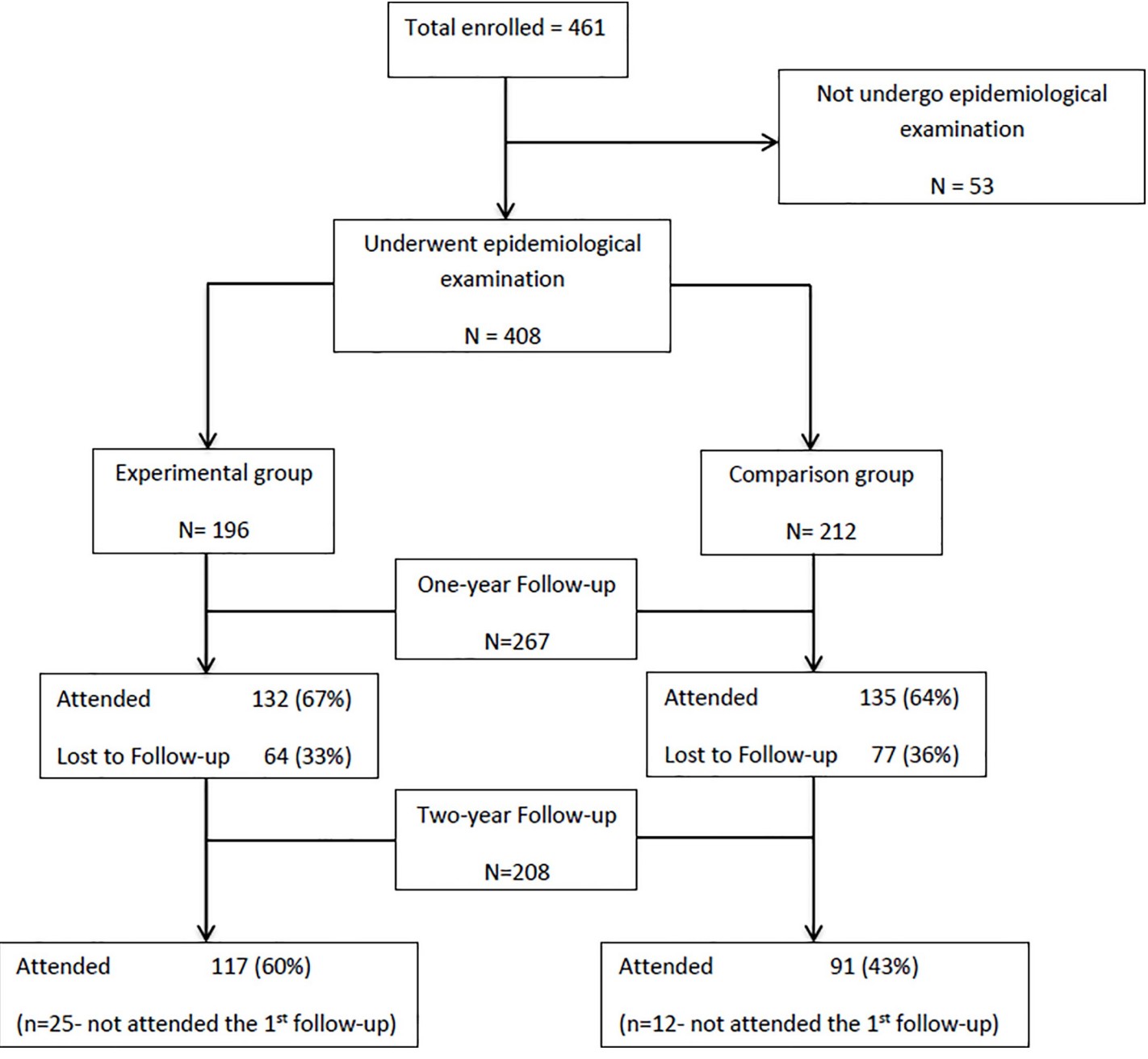

**Fig 1. Sample size and follow-up of children by Intervention and Comparison groups.**

The ITT analysis for caries increment in the permanent dentition demonstrated similar differences to the per protocol analysis, suggesting these results are robust. In particular, the Comparison group experienced greater advanced caries increment in the permanent dentition than the Intervention group. Comparison group children had significantly more advanced carious tooth surfaces in the permanent dentition compared to the Intervention group children (IRR = 1.61, 95%CI 1.20–2.17; p = 0.03) after adjusting for baseline caries experience (S1 Table). This effect remained significant after controlling for all potentially confounding background characteristics (IRR = 1.62; 95% CI: 1.18–2.23; p = 0.03) (S2 Table).

**Table 1. Association between Intervention and Comparison groups and demographic, lifestyle, and oral health characteristics at baseline.** Binary outcomes presented as frequency (percentage). Continuous outcomes presented as mean [Standard Deviation (SD)]. Count outcomes presented as median [Inter-Quartile Range (IQR)].

| | | Intervention group (N = 196) | Comparison group (N = 212) | Significance |
|---|---|---|---|---|
| | | N (%) | N (%) | |
| Sex* | Males | 87 (44.8) | 98 (46.7) | 0.71 |
| | Females | 107 (55.2) | 112 (53.3) | |
| On a typical day, do you drink soft drink* | Yes | 125 (72.3) | 164 (80.8) | 0.05 |
| | No | 48 (27.7) | 39 (19.2) | |
| On a typical day, do you consume fruit juice* | Yes | 162 (92.6) | 193 (94.6) | 0.42 |
| | No | 13 (7.4) | 11 (5.4) | |
| On a typical day, do you eat consume sweets and lollies* | Yes | 146 (84.4) | 180 (88.7) | 0.22 |
| | No | 27 (15.6) | 23 (11.3) | |
| On a typical day, do you consume Syrups, jams and sweet spread* | Yes | 150 (87.7) | 189 (93.1) | 0.07 |
| | No | 21 (12.3) | 14 (6.9) | |
| Do you add sugar to your cereal, tea, coffee or milo* | Yes | 115 (67.3) | 161 (78.9) | 0.01 |
| | No | 56 (32.7) | 43 (21.1) | |
| How often do you brush your teeth* | Once or less than once a day | 38 (22.9) | 55 (27.2) | 0.34 |
| | Twice or more a day | 128 (77.1) | 147 (72.8) | |
| How often do you use toothpaste when brushing your teeth† | Always | 158 (92.4) | 193 (95.1) | 0.41 |
| | Most times | 7 (4.1) | 7 (3.4) | |
| | Sometimes | 6 (3.5) | 3 (1.5) | |
| When was the last time you visited the dental clinic?* | Less than 6 months ago | 23 (26.7) | 12 (15.8) | 0.12 |
| | 6–12 months | 26 (30.2) | 18 (23.7) | |
| | 1 year– 2 years ago | 22 (25.6) | 25 (32.9) | |
| | More than 2 years ago | 15 (17.4) | 21 (27.6) | |
| Salivary pH‡ [Mean (SD)] | | 7.2 (0.4) | 7.07 (0.52) | 0.67 |
| Salivary flow/5 min‡[Mean (SD)] | | 6.15 (3.20) | 5.82 (3.08) | 0.53 |
| Total buffering capacity‡ [Mean (SD)] | | 9.46 (2.02) | 9.1 (2.1) | 0.16 |
| MS levels‡‡ | Low risk | 32 (18.5) | 12 (11.3) | 0.15 |
| | Low risk | 24 (13.9) | 10 (9.4) | |
| | High risk | 39 (22.5) | 23 (21.7) | |
| | High risk | 78 (45.1) | 61 (57.5) | |
| LB levels‡‡ | Low risk | 75 (43.4) | 34 (32.1) | 0.08 |
| | Low risk | 37 (21.4) | 22 (20.8) | |
| | High risk | 39 (22.5) | 25 (23.6) | |
| | High risk | 22 (12.7) | 25 (23.6) | |
| Age§ [Median(IQR)] | | 8 (6–11) | 8 (6–11) | 0.66 |
| Tooth surfaces with incipient caries | | | | |
| Deciduous dentition‡ [Median(IQR)] | | 2 (2–4) | 2 (1–5) | 0.10 |
| Permanent dentition‡ [Median(IQR)] | | 3 (3–8) | 6 (2–13) | <0.001 |
| Tooth surfaces with advanced caries | | | | |
| Deciduous dentition‡ [Median(IQR)] | | 2 (2–5) | 3 (1–8) | 0.03 |
| Permanent dentition‡ [Median(IQR)] | | 0 (0–2) | 1 (0–2) | 0.24 |

*Chi Square analysis

†Chi Square with continuity correction

‡Mann Whitney U test

§Unpaired t-test;

‡‡caries risk based on the MS and LB CFU's was determined based on manufacturer provided model charts (provided in S1 Fig)

Total number of subjects for the variables in the table might not be equal to total sample size due to missing data

**Table 2. Association between group and caries increment at 2-year follow-up: Per-protocol analysis.**

| | Deciduous surfaces | | Permanent surfaces | |
|---|---|---|---|---|
| | Mean (SD) | IRR (95% CI); p-value[a] | Mean (SD) | IRR (95% CI); p-value[a] |
| *Incipient caries increment* | | | | |
| Comparison | 1.75 (2.43) | 1.40 (0.87–2.25); 0.17 | 5.37 (4.47) | 1.16 (0.84–1.62); 0.37 |
| Intervention | 1.23 (1.84) | **Ref** | 4.37 (4.89) | **Ref** |
| Preventive fraction[b] | 30% | | 19% | |
| *Advanced caries increment* | | | | |
| Comparison | 2.00 (2.30) | 1.10 (0.68–1.77); 0.70 | 0.96 (1.44) | 1.61 (1.02–2.54); **0.04** |
| Intervention | 1.53 (2.64) | **Ref** | 0.55 (1.31) | **Ref** |
| Preventive fraction[b] | 24% | | 43% | |
| *Total caries increment* | | | | |
| Comparison | 3.75 (3.87) | 1.25 (0.93–1.68); 0.15 | 6.33 (4.75) | 1.26 (0.88–1.80); 0.21 |
| Intervention | 2.76 (3.63) | **Ref** | 4.91 (5.30) | **Ref** |
| Preventive fraction[b] | 26% | | 22% | |

[a]Negative binomial with log link regression adjusted for baseline caries experience, P<0.05 in bold font

[b]Preventive fraction = mean increment in Comparison–mean increment in Intervention ÷ mean increment in Comparison X 100

## Discussion

The intervention to treat all existing caries at baseline followed by a 'Big Bang' annual application of selective fissure sealants, povidone-iodine and fluoride varnish reduced the number of new tooth surfaces with advanced caries on the permanent dentition in the Intervention group compared to the Comparison group. At the 2-year follow-up, the caries preventive fraction due to the intervention ranged between 20% to 40% for all caries outcomes.

Direct comparison of our findings to literature is difficult as investigation of annual application of interventions of this type, to our knowledge, does not exist. The three types of intervention applied have each been reported to individually have significant success in preventing dental caries [15–17]. However, a recent report from a rural non-fluoridated community in Chile found bi-annual fluoride application not effective in preschool children [18]. A community-randomised controlled trial in Australian Aboriginal children did show that fluoride varnish was efficacious [19]: in this trial caries reduction was 2.3–3.5 surfaces/child with preventive fraction of 24–36%; similar to our findings. A trial in a remote Australian Indigenous community measuring the effectiveness of sliver fluoride for deciduous-dentition caries showed it as effective as the Atraumatic Restorative Technique (ART) [20].

While our annual intervention reduced caries incidence, caries increment remains unacceptably high in this community, even in children who received it. These interventions are resource-intensive, so consideration should be given to proven cost-effective interventions such as the reintroduction of water fluoridation. If active interventions are needed, consideration should be given to expanding the roles of community health workers to deliver preventive programs, especially minimally invasive interventions such as povidone-iodine applications and fluoride varnish. In remote communities it would not be feasible or sustainable for these to be delivered by oral health professionals. If performed by trained community health workers, more regular applications as normally recommended would be feasible [21].

Besides introducing active (professionally-applied) and passive (water fluoridation) interventions, it is of critical importance to address social determinants to reduce the burden of poor oral health [22]. For example, consumption of sugar-sweetened beverages is of serious

**Table 3. Multivariable analyses of advanced caries increment in the permanent dentition at 2-year follow-up as the dependent variable (per-protocol)–adjusted analysis.**

| Explanatory variables | IRR (95% CI) | P |
|---|---|---|
| **Group allocation** | | |
| Comparison | 2.21 (1.03–4.71) | **0.04** |
| Intervention | **Ref** | |
| **Sex** | | |
| Males | 0.72 (0.33–1.55) | 0.396 |
| Females | **Ref** | |
| **Age (per year)** | 1.24 (1.07–1.44) | **<0.01** |
| **Baseline caries experience (per surface)** | 1.11 (0.97–1.27) | 0.143 |
| **Brushing group** | | |
| Once or less | 1.02 (0.46–2.26) | 0.97 |
| Twice or more | **Ref** | |
| **Soft drinks consumption on a typical day** | | |
| Yes | 1.56 (0.52–4.64) | 0.43 |
| No | **Ref** | |
| **Sweets on Lollies consumption on a typical day** | | |
| Yes | 1.06 (0.29–3.83) | 0.93 |
| No | **Ref** | |
| **Syrups and Jams consumption on a typical day** | | |
| Yes | 1.35 (0.36–5.03) | 0.65 |
| No | **Ref** | |
| **Adding sugar to cereal, tea, coffee or milo on a typical day** | | |
| Yes | 0.86 (0.30–2.45) | 0.78 |
| No | **Ref** | |
| **Salivary Mutans Streptococci** | | |
| Low risk | 0.31 (0.11–0.88) | **0.03** |
| High risk | **Ref** | |
| **Salivary LB levels** | | |
| Low risk | 0.49 (0.24–0.99) | **0.05** |
| High risk | **Ref** | |

P<0.05 in bold font

concern in Indigenous communities, with increases occurring during adolescence [23]. Strategies to reduce sugar consumption include taxes on sugar and/or sugar-sweetened beverages [24], and graphic warnings on labels [25]. It is incumbent upon dental researchers, dental providers and policy-makers to advocate for public health interventions to address the upstream social determinants of health that have the most impact on the oral and general health [26].

At this is a non-randomised trial, lacking a prospective randomised recruitment, assessing the true effect of the preventive intervention is compromised. The two groups for example showed pre-existing differences at baseline. Longitudinal studies, especially in remote settings, have limitations. One such is loss to follow-up of participants. In our community there are a number of reasons: obtaining parent/guardian consents is always challenging, not necessarily because there is opposition, often simply because carers cannot be located or motivated. Here, this was compounded by the need to obtain multiple signed consents: for epidemiological

examinations; for application of our preventative intervention, for treatment planning examinations and for restorative treatments. Similarly, compliance to all stages of clinical contact is often poor, because students are absent from school or out of the community for social or family reasons. School absenteeism is common and it is an important goal of government to improve this [27]. In 2015, at baseline, school attendance was high as there was a community effort to encourage this. At follow-up visits this effort had waned and absenteeism was higher. Some children, especially later into their schooling, move to larger towns to complete their education. While we lost a number of children to follow up over the 2-years of the study, the findings from the per-protocol and ITT analysis showed no differences. While clinical examiners were not informed of the group status of children who attended follow-up, the presence of fissure sealants made blinding impossible. With each examination time point separated by a year, deciduous teeth that developed caries, but exfoliated and were replaced by permanent teeth between these time points would have underestimated caries increment in the deciduous dentition.

While it is important to address the social determinants of health in Indigenous communities in Australia, it is critically important that at a national level there is progress on the broader issues of Indigenous disadvantage and dispossession of land and resources. Many argue for formal recognition of the Aboriginal and Torres Strait Islander peoples in the Nation's constitution, for a formal treaty to acknowledge the impact of colonisation and a Makarrata: *the coming together after a struggle* [28], all of which impact on oral health, oral health-related quality of life and overall health and well-being.

## Conclusion

Remote Indigenous children who received the annual preventive intervention experienced fewer incident advanced carious lesions in the permanent dentition at the 2-year follow-up, compared to children who did not receive this.

## Supporting information

**S1 Checklist.**
(DOC)

**S1 Fig.**
(TIF)

**S1 Table. ITT analysis of caries increment at two-year follow-up between Intervention and Comparison groups.**
(DOCX)

**S2 Table. Multivariable analyses with increment in advanced caries at two-years follow-up as the dependent variable (ITT)–adjusted analysis.**
(DOCX)

**S1 File.**
(PDF)

**S2 File.**
(PDF)

**S3 File.**
(PDF)

## Acknowledgments

We pay our respects to the traditional owners of the lands on which this research took place: the Anggamuthi, Atambay, Wuthathi, Yadhaykenu and Gudang peoples of the Northern Peninsula Area (NPA); the Kaurna people of the Adelaide Plains; the Yugambeh/Kombumerri peoples of the Gold Coast; and the Turrbal and Jagara people of the Brisbane (Meanjin) area. The authors gratefully acknowledge the Elders, Community Members & Community Workers in the Northern Peninsula Area of Far North Queensland and the Principals, Staff & Children of the Northern Peninsula Area State College. Our sincerest thanks to all Chief and Associate Investigators and Project Managers.

## Author Contributions

**Conceptualization:** Ratilal Lalloo, Jeroen Kroon, Newell W. Johnson.

**Data curation:** Ratilal Lalloo, Santosh K. Tadakamadla, Newell W. Johnson.

**Formal analysis:** Ratilal Lalloo, Santosh K. Tadakamadla, Jeroen Kroon, Lisa M. Jamieson, Robert S. Ware, Newell W. Johnson.

**Funding acquisition:** Jeroen Kroon, Newell W. Johnson.

**Investigation:** Jeroen Kroon, Newell W. Johnson.

**Methodology:** Ratilal Lalloo, Jeroen Kroon, Lisa M. Jamieson, Newell W. Johnson.

**Project administration:** Jeroen Kroon, Newell W. Johnson.

**Resources:** Newell W. Johnson.

**Supervision:** Newell W. Johnson.

**Validation:** Newell W. Johnson.

**Visualization:** Newell W. Johnson.

**Writing – original draft:** Ratilal Lalloo, Newell W. Johnson.

**Writing – review & editing:** Ratilal Lalloo, Santosh K. Tadakamadla, Jeroen Kroon, Lisa M. Jamieson, Robert S. Ware, Newell W. Johnson.

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
