## [Decision Letter · Decision Letter 0]

1 Nov 2019

PONE-D-19-20492

Impact of a non-randomised caries preventive trial in remote Indigenous Australian children

PLOS ONE

Dear Prof Johnson,

The manuscript has been assessed by three reviewers; their comments are available below.

The reviewers find the subject of relevance but two of them have raised major methodological concerns about the study. The reviewers note concerns about the matching between groups and note that the fact that participants self selected into the intervention group may have introduced a major confounder, the reviewers also request sociodemographic information about the participants. The reviewers raise major concerns about the statistical analyses reported and note that the data should be re-analysed using different approaches.

Could you please carefully revise the manuscript to address the concerns raised by the reviewers and submit a revised version of the manuscript for further consideration.

Please note that the revised manuscript will need to undergo further review, we thus cannot at this point anticipate the outcome of the evaluation process.

We would appreciate receiving your revised manuscript by Dec 15 2019 11:59PM. Please include the following items when submitting your revised manuscript:

We look forward to receiving your revised manuscript.

Kind regards,

Iratxe Puebla

Senior Managing Editor, PLOS ONE

Journal Requirements:

2. We note that you have provided a copy of the published clinical trial protocol (BMC 2015). Note that PLOS ONE requires authors to provide a copy of the original clinical trial protocol that was submitted to and approved by the ethics committee, as well as any modifications. Please update your supporting information files to include this documentation.

3. Also, please provide additional details regarding participant consent.

In the ethics statement in the Methods and online submission information, please ensure that you have specified (a) whether consent was informed and (b) what type you obtained (for instance, written or verbal, and if verbal, how it was documented and witnessed).

Please ensure you have stated whether you obtained consent from parents or guardians for all included participants.

4. We noted in your submission details that a portion of your manuscript may have been presented or published elsewhere:

'An earlier version appears at:

Impact of a caries preventive intervention in remote Indigenous Australian children

R Lalloo, SK Tadakamadla, J Kroon, LM Jamieson, NW Johnson

bioRxiv, 585935'

Please clarify whether this publication was peer-reviewed and formally published. If this work was previously peer-reviewed and published, in the cover letter please provide the reason that this work does not constitute dual publication and should be included in the current manuscript.

Reviewers' comments:

Reviewer's Responses to Questions

**Comments to the Author**

1. Is the manuscript technically sound, and do the data support the conclusions?

Reviewer #1: Partly

Reviewer #2: Partly

Reviewer #3: Yes

2. Has the statistical analysis been performed appropriately and rigorously? 

Reviewer #1: No

Reviewer #2: No

Reviewer #3: Yes

3. Have the authors made all data underlying the findings in their manuscript fully available?

Reviewer #1: No

Reviewer #2: No

Reviewer #3: Yes

4. Is the manuscript presented in an intelligible fashion and written in standard English?

Reviewer #1: Yes

Reviewer #2: Yes

Reviewer #3: Yes

5. Review Comments to the Author

Reviewer #1: The manuscript addresses an interesting topic. The research questions are well-posed and the data contain a rich set of information. The employed statistical methods are very basic and not all the data features have been accounted properly. Some comments follow.

1. The data are not fully available. This is not in line with the journal's guidelines. Moreover, this does not allow the reviewer to reproduce the results. This is particularly important as several sentence must be verified, e.g. the fact the mathcing is properly performed,"largely matched the characteristics of the comparison group" is quite a strong sentence if not supported by any evidence.

2. As missing data arise, more details on the multiple imputation procedure should be provided, pointing out the potential bias and/or increase in the uncertainty (e.g. standard errors of the estimates) due to the imputation approach. As a general matter, an analysis of the imputed data should be provided. It is important to inspect the imputations. In general, a good imputed value is a value that could have been observed had it not been missing.

3. A major concern refers to the analysis of the main outcome. The ICDAS classification leads to a categorical ordered variable. The use of a non-parametric test is sound, though reporting the "mean" is rather misleading. Moreover, the data are rich of information. A longitudinal data structure is available and completely neglected. A mixed effects model must be employed to better appreciate the differences between the groups, taking into account possible individual-specific confounders and a possible trend/growth effect. Heterogeneity may be quite an issue, in particular in small/medium sample size analyses as few observation may strongly affect inferential procedures.

4. As a minor point, it is rather unclear why the Permanent surface and All surfaces values differ as they are computed on the same number of subject, with the All surfaces values that are often greater than the others.

Reviewer #2: Review attached

Please use the space provided to explain your answers to the questions above. You may also include additional comments for the author, including concerns about dual publication, research ethics, or publication ethics. (Please upload your review as an attachment if it exceeds 20,000 characters) (Limit 200 to 20000 Characters)

Reviewer #3: I would like to congratulate the authors and declare that I am very thankful for the opportunity to revise this manuscript. It is well designed, conducted and reported. This topic is extremely important to improve the public health politic for disadvantaged populations. However, some points should be improved:

Previous studies have showed that initial noncavitated caries lesion have no higher risk of developing a severe increment in caries. Studies have also showed that many of these lesion arrest without a interevention by dentist. In this cases, the preventive politics should be considered in order to avoid frank dentin cavitation. Thus, the primary outcome should be focused in more severe caries lesion. Discussion should also be focused in results about moderate-advanced caries lesions.

Furtheremore, the conclusion should be based only in the results. Speculations or ideas should be added in discussion section.

6. PLOS authors have the option to publish the peer review history of their article (what does this mean?). If published, this will include your full peer review and any attached files.

Reviewer #1: No

Reviewer #2: No

Reviewer #3: Yes: Tamara Kerber Tedesco

---

## [Author Response · Author response to Decision Letter 0]

11 Mar 2020

Reviewer 1: Responses:

The manuscript addresses an interesting topic. The research questions are well-posed and the data contain a rich set of information. The employed statistical methods are very basic and not all the data features have been accounted properly. Some comments follow. The full statistical analysis has been re-worked, based on detailed comments from both reviewers. Additional details of the changes are provided with the relevant comments.

In summary the statistical analysis now focused on the 2-year follow-up, and the 1-year follow-up findings have been removed. The all surface (combination of two dentitions) have also been removed to avoid confusion. 

We have also removed the comparison of experimental group children who received one and two preventive intervention over the study period as this is also possibly confusing but also adds little value to the overall analyses conducted. 

We have now included the baseline comparison findings for the two groups.

We have reported Medians (95% Confidence intervals) instead of Means (SDs).

We have conducted and reported the per-protocol multivariate analysis to assess the effect of the intervention after adjusting for potential confounders, focusing on the important outcome (as suggested by reviewer 3) of advanced caries increment in the permanent dentition. 

We have replicated the per-protocol analysis on Intention to Treat (ITT) basis for both bivariate and multivariate analyses.

The data are not fully available. This is not in line with the journal's guidelines. Moreover, this does not allow the reviewer to reproduce the results. This is particularly important as several sentence must be verified, e.g. the fact the mathcing is properly performed,"largely matched the characteristics of the comparison group" is quite a strong sentence if not supported by any evidence. Access to raw data will be considered on request. Because of cultural sensitivities these will be referred to the Griffith University Research Ethics Committee and the Community will be contacted for advice. 

A detailed table presenting the baseline comparison of the two groups is now included with supporting text in the Results section. 

As missing data arise, more details on the multiple imputation procedure should be provided, pointing out the potential bias and/or increase in the uncertainty (e.g. standard errors of the estimates) due to the imputation approach. As a general matter, an analysis of the imputed data should be provided. It is important to inspect the imputations. In general, a good imputed value is a value that could have been observed had it not been missing. Data from the imputed analysis are now included in the revised manuscript. Standard errors are presented in the multivariate analysis tables. 

A major concern refers to the analysis of the main outcome. The ICDAS classification leads to a categorical ordered variable. The use of a non-parametric test is sound, though reporting the "mean" is rather misleading. Moreover, the data are rich of information. A longitudinal data structure is available and completely neglected. A mixed effects model must be employed to better appreciate the differences between the groups, taking into account possible individual-specific confounders and a possible trend/growth effect. Heterogeneity may be quite an issue, in particular in small/medium sample size analyses as few observation may strongly affect inferential procedures. The outcome in this study was the incidence of new lesions, i.e., the count of incipient and advanced caries which developed on tooth surfaces that were caries free at baseline. Means were reported as it is easy to interpret. We have now reported medians with 95% CIs, as recommended. 

We have discussed conducting a mixed effects model with our resident statistics Professor who had some reservations in relation to use of mixed effects model on this data where the outcome is incidence (new lesions). A mixed effects model would undermine the use of valid incidence data by only considering the prevalence of dental caries at each timepoint. We have now conducted a negative binomial regression with log link and have taken into account potential confounders including baseline caries as suggested. 

As a minor point, it is rather unclear why the Permanent surface and All surfaces values differ as they are computed on the same number of subject, with the All surfaces values that are often greater than the others. Apologies, if we did not present this adequately. Yes, all surface values are greater than permanent surfaces as ‘all surfaces” is a sum of both deciduous and permanent surfaces. The mean of ‘all surfaces’ is dependent on deciduous and permanent surfaces but is not the total sum as the number of individuals with deciduous and permanent dentitions is not the same.

We have however removed the all surface findings to avoid any confusion.

Reviewer 2: Responses:

I would like to thank the authors and the editors for the chance to review this manuscript. Dental caries in high risk, hard to reach populations is a critical research area and one that this reviewer has particular interest in. As such, rigorous research is a welcome contribution. We thank the referee for his understanding and support 

The authors report on a non-randomized study of the combined effects of fluoride varnish and pit/fissure sealants for the prevention of dental caries in indigenous populations. The individual effectiveness of these agents in the prevention of caries is well known and has been thoroughly studied, as well as being formally recommended by multiple national agents for this purpose. The authors argue that the combined effects of these agents have not been evaluated, but I believe there are studies currently in operation that have still been studied. Some have investigated this combination both in clinical and pragmatic studies, though in varying forms (e.g., 10.1186/s12903- 018-0514-6; 10.1002/14651858.CD003067.pub4; 10.5005/jp-journals-10005-1491; 10.1016/j.adaj.2017.05.028), which may be useful to cite in the literature of the study. This reviewer knows also of two ongoing clinical trials looking at the impact of the combination of therapies, one arm of which is FV+sealants. The Introduction has been updated to include the suggested research evidence.

That said, there is still need for further evidence of the combined effects of these agents in high-risk alternative populations. Thus, the overall scientific contribution from this type of research can be positive. The authors do a wonderful job of presenting their study in detail. However, there are substantial methodological concerns as described below that weakened enthusiasm for this manuscript. 

This reviewer disagrees with non-consenting children being considered a “natural” comparison group. They can be treated as a comparator but this language is, I believe, too confusing and similar to that of natural experiments. Further, I was unable to find a comparison (baseline or otherwise) between the consenting and non-consenting children. The obvious concern is that there are systematic differences in patients that consent and do not consent to the study. In fact, there seem to be no tables or results anywhere regarding the sociodemographic breakdown of the participants in either group. There is a sentence in results saying they “largely matched”, but there is no table and it needs to be shown. A detailed table presenting the baseline comparison of the two groups is now included with supporting text in the Results section.

The outcome of caries incidence is complicated by exfoliation. The age group on the low end means there may have been instances wherein primary teeth were exfoliated during the study and should this data have been carious, it is biased. 

 When caries incidence was computed only those missing due to caries were considered. i.e., we used ICDAS code 97. Examiners inquired if the tooth was missing due to caries. However, it is possible that inaccuracies can arise if deciduous teeth developed caries but exfoliate and are replaced by a permanent tooth between the examination time points. We have now included a statement on this as a limitation.

I am confused by a sentence in results about extraction due to caries, I thought this was a program due to a lack of care access for the patient population (typical for this type of intervention). Yet children with carious lesions have access to extractions? Could the authors clarify?

 There is a public dental service available to the community, albeit limited, in the community and a distant island, only accessible by ferry. These services are however essentially for adults in the community, although children are treated, usually in emergencies. The number of teeth extracted during the study period should be small and not impact on the findings of the study. An accurate account of services received outside of the study would require a complex and lengthy (up to six to 12 months) approval process to receive. There is no private dental service in this community or nearby. 

Post hoc sample size calculation for power is generally not recommended. 

 Thanks for recommendation, we have deleted this information. 

As the intervention was not randomized and the children effectively self-selected into the study intervention and comparator groups, it is unclear if the effect is true. E.g., if consenting children have greater wealth (as a simple example) or more attentive parents, the home environment may be more conducive to a healthier diet and/or hygiene behavior. This is probably the most problematic limitation. 

 A detailed table presenting the baseline comparison of the two groups is now included with supporting text in the Results section.

Please justify the MAR assumption.

 We performed Littles MCAR test with the p-values being significant for all the variables with the missing data. With the p-values being significant, the missing data could not be MCAR. We also compared the baseline characteristics between the dropouts and those completing the experiment to assess the dropout bias. There was no dropout bias and we cannot conceptualize any unobserved variables that could predict the missingness of the data in this population, particularly the outcome data. Therefore, MNAR is out of question. 

This leaves us with the assumption that the data is MAR. With the assumption of MAR, we have included all the variables that could be theoretically predictive of missing data while imputing the missing data as can be seen in the description in the statistical analyses. This is commonly accepted practice. 

Sterne JA, White IR, Carlin JB, Spratt M, Royston P, Kenward MG, Wood AM, Carpenter JR. Multiple imputation for missing data in epidemiological and clinical research: potential and pitfalls. BMJ. 2009 Jun 29;338:b2393. doi: 10.1136/bmj.b2393.

https://www.ncbi.nlm.nih.gov/pubmed/19564179

Dong Y, Peng CY. Principled missing data methods for researchers. Springerplus. 2013 May 14;2(1):222. doi: 10.1186/2193-1801-2-222.

https://www.ncbi.nlm.nih.gov/pubmed/23853744

The MW test is a fairly basic nonparametric test. Given the non-randomization of the design this can lead to serious concerns over confounding, and the authors did not seem to do any adjustment (e.g., propensity score matching or inverse probability weighting) to alleviate this concern. 

 There is a plethora of literature supporting the use of regression analysis controlling the effect of confounding variables over matching. Also, matching is not as sensitive to the functional form of covariates as in regression. Therefore, we used multivariate regression to control the confounding. 

Hettle R, Corbett M, Hinde S, et al. The assessment and appraisal of regenerative medicines and cell therapy products: an exploration of methods for review, economic evaluation and appraisal. 

https://www.ncbi.nlm.nih.gov/books/NBK424728/

Brazauskas R, Logan BR. Observational Studies: Matching or Regression? Biol Blood Marrow Transplant. 2016 Mar;22(3):557-63. doi: 10.1016/j.bbmt.2015.12.005. 

https://www.ncbi.nlm.nih.gov/pmc/articles/PMC4756459/

Alemayehu D, Alvir JM, Jones B, Willke RJ. Statistical issues with the analysis of nonrandomized studies in comparative effectiveness research. J Manag Care Pharm. 2011 Nov-Dec;17(9 Suppl A):S22-6. 

https://www.ncbi.nlm.nih.gov/pubmed/22074671

We have included the results from negative binomial regression analysis with log link. Adjustments were made for potential confounding variables including baseline caries experience. 

The reviewer cannot find tables on the baseline measures for caries between groups. A detailed table presenting the baseline comparison of the two groups is now included with supporting text in the Results section.

There are a number of MWU tests performed here, yet no treatment or discussion is given to the possibly large experiment wise error rate. The authors should consider nonparametric regression or multivariate regression of transformed outcomes, which would also allow them to consider the varying sociodemographic variables in the model. Thanks for the suggestion. We have conducted a negative binomial regression analysis with log link with caries increment as the outcome variable. 

ITT and PPA was conducted, and the authors went with PPA, yet no justification is given. Due to the analyses being similar and due to space limitation, we opted to only present the PPA findings in the original submission. However, we have now presented both the PPA and ITT findings for the bivariate and multivariate analyses. 

Results: please provide confidence intervals. The authors note the non-normality of outcome data yet did not describe the severity of this departure. Given the MW test, the authors could use a Wilcoxon test CI for medians or log-transform their data and calculate a modified Cox confidence interval. These will be more useful than with P-values. We have now provided medians with 95% CIs. 

The authors should be careful of the use of “effective” and “effectiveness” given the nature of the design of the study. We have used the word “efficacious” where appropriate.

In the Discussion, it is confusing to read a paragraph about SSBs. This reviewer understands that SSBs are a significant contributor to caries, but the authors’ reversal here to focus on social determinants when the paper was focused on care provision seems out of place. This is a small component of the Discussion, and we believe that emphasizing addressing the social determinants of dental caries is important; and have therefore retained this in the Discussion. 

One of the concerns in this report is the authors’ own admission that while sealants can be a “set it and forget it” intervention, general clinical recommendations include the multiple reapplication of fluoride varnish. The authors admit, understandably, that this is infeasible for the specific population, however this creates an issue with the interpretation of the results. It is unable to be identified whether the major preventive effect on the observed incidence of caries is due to the longer-lasting sealants, an atypical response to a single application of FV, or a combination of both. Ideally, this study would have had additional treatment groups for just FV and just sealants, but this absence means there is a notable limitation in the impact of the paper. It is not ethical to create another group for just FV and FS when it is evident that both are effective. The study would probably not have been funded and receive ethics approval if such a design was planned. This study design was planned with consideration to the sensitivities for conducting research in a remote Indigenous community. We have added:

“ it must be emphasized that the purpose of the study was to determine the efficacy, and cost effectiveness, of professional interventions known to be effective individually, but deliberately applied together for maximum effect. The study is essentially a piece of health economics research: it was not designed to evaluate disinfectants, FV and sealants individually”.

Overall, despite the nice design and excellent presentation of the manuscript, the methodological limitations lead the reviewer to believe it is not of a high enough standard for publication in PLOS One. As a non-randomized trial, one would expect to see more rigorous treatment of internal and external validity threats in analysis, but the (over)use of MWU tests was considered weak. Additionally, the self-selection of participants into treatment and control was also considered a major limitation. We have re-worked the full statistical analysis. Self-selection of participants is tied with the ethical approval and consent processes of the study.

Research in remote Indigenous communities is challenging. Whilst we had strong support from Elders and local school and health care personnel, it is difficult to reach and engage every single family. A degree of pragmatism is required. We are convinced that we have made a positive impact on the health and wellbeing of these children and this is supported by our econ omics evaluation, currently in press.

HQLO-D-19-00484R1

Demonstration of high value care to improve oral health of a remote Indigenous community in Australia Sanjeewa Kularatna, PhD; Ratilal Lalloo, PhD; Jeroen Kroon; Santosh K K Tadakamadla, PhD; Paul A scuffham, PhD; Newell W Johnson, PhD Health and Quality of Life Outcomes

Criteria notes:

- The results are original research and have not been published elsewhere. 

- Overall, the article is intelligible and well-written.

- The study provided acceptable ethical approval

- The paper follows standard reporting guidelines 

Reviewer 3:

Responses:

I would like to congratulate the authors and declare that I am very thankful for the opportunity to revise this manuscript. It is well designed, conducted and reported. This topic is extremely important to improve the public health politic for disadvantaged populations. However, some points should be improved: 

Previous studies have showed that initial noncavitated caries lesion have no higher risk of developing a severe increment in caries. Studies have also showed that many of these lesion arrest without a interevention by dentist. In this cases, the preventive politics should be considered in order to avoid frank dentin cavitation. Thus, the primary outcome should be focused in more severe caries lesion. Discussion should also be focused in results about moderate-advanced caries lesions. Well OK, and the res.t............

 We do separate incipient and advance caries increment in the analysis. The multivariate model only analyses advances caries increment in the permanent dentition.

Furtheremore, the conclusion should be based only in the results. Speculations or ideas should be added in discussion section. The conclusion has been edited to focus on the findings, and speculations and ideas have been moved to the Discussion.

---

## [Decision Letter · Decision Letter 1]

16 Jun 2020

PONE-D-19-20492R1

Impact of a non-randomised caries preventive trial in remote Indigenous Australian children

PLOS ONE

Dear Dr. Johnson,

Thank you for submitting your manuscript to PLOS ONE. After careful consideration, we feel that it has merit but does not fully meet PLOS ONE’s publication criteria as it currently stands. Therefore, we invite you to submit a revised version of the manuscript that addresses the points raised during the review process.

I apologize for the amount of time that has elapsed since you initially submitted this manuscript. We have since obtained further comments from the 3 original reviewers, whose comments are available below. As you will read, these individuals have a number of concerns regarding the statistical analyses used in your trial. Can you please carefully revise your manuscript to address the comments of the reviewers? We ask that you please pay close attention to the comments by reviewer #1. 

We look forward to receiving your revised manuscript.

Kind regards,

Anna Simonin, Ph.D.

Senior Editor

PLOS ONE

Reviewers' comments:

Reviewer's Responses to Questions

**Comments to the Author**

1. If the authors have adequately addressed your comments raised in a previous round of review and you feel that this manuscript is now acceptable for publication, you may indicate that here to bypass the “Comments to the Author” section, enter your conflict of interest statement in the “Confidential to Editor” section, and submit your "Accept" recommendation.

Reviewer #1: (No Response)

Reviewer #2: (No Response)

Reviewer #3: All comments have been addressed

2. Is the manuscript technically sound, and do the data support the conclusions?

Reviewer #1: Yes

Reviewer #2: Partly

Reviewer #3: Yes

3. Has the statistical analysis been performed appropriately and rigorously? 

Reviewer #1: Yes

Reviewer #2: No

Reviewer #3: Yes

4. Have the authors made all data underlying the findings in their manuscript fully available?

Reviewer #1: Yes

Reviewer #2: Yes

Reviewer #3: Yes

5. Is the manuscript presented in an intelligible fashion and written in standard English?

Reviewer #1: Yes

Reviewer #2: Yes

Reviewer #3: Yes

6. Review Comments to the Author

Reviewer #1: I thank the authors for the replies to all my comments.

I have two minor points to be further discussed.

a. Some sensitivity analysis about the imputation data used must be included. How many imputed datasets did you consider? Of course, several must be considered.

b. I appreciate the use of the negative binomial regression model. I feel that as you are modelling the new lesions, the inclusion of random effects is necessary to account for the longitudinal strucutre of the data. The book from Rainer Winkelmann "Econometric Analysis of Count Data" could be a good reference to get why random effects may be used. Of course, please provide the results of the likelihood ratio test to check if the random effects are needed or not.

Reviewer #2: (No Response)

Reviewer #3: The article is ready to be published and will bring relevant information about poorly considered populations.

7. PLOS authors have the option to publish the peer review history of their article (what does this mean?). If published, this will include your full peer review and any attached files.

Reviewer #1: No

Reviewer #2: No

Reviewer #3: Yes: Tamara Kerber Tedesco

---

## [Author Response · Author response to Decision Letter 1]

9 Sep 2020

Our sincerest thank you to all the reviewers for their comments on our resubmission. Below we summarise our responses. Substantial changes based on the comments are highlighted in yellow in the revised manuscript.

Reviewer 1:

Comment: I have two minor points to be further discussed.

a. Some sensitivity analysis about the imputation data used must be included. How many imputed datasets did you consider? Of course, several must be considered.

Response: This is included in the methods under the statistical analysis section, lines 157-158 on page 8 - ten iterations were used, and five imputed datasets were obtained.

Comment: b. I appreciate the use of the negative binomial regression model. I feel that as you are modelling the new lesions, the inclusion of random effects is necessary to account for the longitudinal strucutre of the data. The book from Rainer Winkelmann "Econometric Analysis of Count Data" could be a good reference to get why random effects may be used. Of course, please provide the results of the likelihood ratio test to check if the random effects are needed or not.

Response: Thank you for your suggestion. To account for the repeated measures nature of the data we have adjusted all regression models for caries experience at baseline. This has the effect on including both measurements of the outcome under investigation in the same model.

Lines 144-153, page 7:

“The association between group and caries increment was investigated using negative binomial regression models. In the initial model, group was included as the main effect and baseline caries experience as the covariable. Multivariable analyses with full adjustment for all covariables were conducted for the caries outcomes that demonstrated significant associations with group in the initial model. Covariables included in the adjusted models were age, sex, baseline caries experience, soft drink, lolly, syrup/jam, sugar consumption, salivary mutans streptococci and salivary LB levels. The negative binomial model was chosen due to the overdispersion of the outcome data, and the consequent inappropriateness of a Poisson model for this count data. Effect estimates are presented as Incidence Rate Ratios (IRR) with 95% confidence intervals (CIs).”

Reviewer 2:

Comment: In the introduction, the authors again used “efficacy” in one sentence and “effectiveness” in another. These are different concepts and, one could argue, both incorrect for the study as presented. Later on they say “impact”, when really all they can say in this study is association. 

Response: In the Introduction we have replaced ‘efficacy’ with association and “effectiveness” with impact. Effectiveness appears elsewhere in the manuscript, but these relate to the evidence reviewed.

Comment: The nonconsenting children do not constitute a “natural comparison group”. This needs to be changed, and was mentioned in my previous review. 

Response: The word natural has been removed, the non-intervention group is now referred to as the Comparison group. 

Comment: It is a bit of a misnomer to call the treatment group the “experimental group”. It not as egregious as natural comparison but they should consider a different name. This is not an RCT. 

Response: We have changed the name of the “experimental group” to the Intervention group. 

Comment: This reviewer does not agree with the authors responses regarding their approach to confounding. First and foremost, I believe they are conflating typical regression adjustment for confounding with causal effects, particularly in their included citation of regression versus matching. Respectfully, that is not a statistical paper. Matching is meant to try to eliminate the potential selection bias by identifying all possible variables associated with the probability of treatment assignment, given the non-random assignment of the participants of the study. I would suggest the reviewers review Morgan & Winship, “Counterfactuals and Causal Inference”, as well as Hernan and Robins, “Causal Inference”, for a thorough discussion regarding—and comparison of—matching, regression adjustment, and weighting in observational research. There are interesting chapters on each that would greatly help the authors in constructing the rationale for their methods and for conducting the analyses. I believe you will find, generally, that when simply using covariate adjustment the results are not as robust as another method (e.g., IPW-MSM, PSM, IV estimation, etc.)

Response: Following the reviewer’s suggestions we have refined the language in this manuscript to emphasize that we are measuring the association between children who received the intervention and children who did not in terms of their caries experience, and are not attempting to make any claims regarding causal effects. For key outcomes we have presented both unadjusted effect estimates, and also effect estimates adjusted for a number of potentially confounding variables. We have interpreted the resulting effect estimate (in this case an incidence rate ratio) in the standard manner – it represents the difference in incidence rates between the intervention and comparison groups when all covariables in the model are held constant. That is, it is a measure of the association between the intervention and comparison groups after adjusting for a string of potentially confounding variables

Comment: As a minor comment, multivariate regression is multiple outcomes. The authors more likely mean multiple regression (or less commonly, multivariable but that tends to only lead to greater confusion). This needs to be changed. 

Response: Thank you. We agree and have changed to the more correct term “multivariable”.

Comment: While there is now a table 1 included in the manuscript, this is of course just the distribution of observable characteristics, and the potential flaw of any cross-sectional (or observational study in general) is that unobservable characteristics still confound the relationship in your study. 

Response: In any observational study there are a large number of unmeasured characteristics. These characteristics may, or may not, be balanced between groups (of course, without measuring them we cannot know if they are balanced or not). It is important to note that the association between groups and caries remained of the same magnitude after adjusting for potentially confounding variables, and that these variables included demographic characteristics, sugar consumption behavior, and baseline caries experience. Extensive literature reviews inform us that these are the key variables that need to be accounted for when adjusting regression models. Consequently, while it is possible unobserved confounders exist, we believe it is unlikely there are other unmeasured confounders that would significantly alter the magnitude of the associations reported in this manuscript.

Comment: Table 2 needs to have the 2-year comparison in the title

Response: Revised - Table 2: Association between group and caries increment at 2-year follow-up: per-protocol analysis.

Comment: If I understand correctly, at baseline there were differences in teeth with incipient caries between treatment and control groups for permanent dentition and advanced caries for deciduous teeth. The authors then report significant differences at the end of the second follow-up in incipient caries increment of permanent surfaces. It does not seem that they did (or were able) to control for baseline differences. This would mean (again, assuming I read correctly) that the results in Table 2 are invalid. 

Response: It is true that there were significant differences between the intervention and comparison groups for incipient caries in the permanent dentition and advanced caries in the deciduous dentition. As suggested, we have therefore now conducted negative binomial regressions adjusted for ‘baseline caries experience’ for results in table 2 followed by a fully adjusted multivariable analysis in table 3. Also, it is to be noted that advanced caries in the permanent dentition was the only outcome that was significantly associated with the group allocation as can be seen in table 2. Therefore, a fully adjusted multivariable analysis was only conducted for this outcome as can be seen in table 3.

Lines 144-153, page 7: 

The association between group and caries increment was investigated using negative binomial regression models. In the initial model, group was included as the main effect and baseline caries experience as the covariable. Multivariable analyses with full adjustment for all covariables were conducted for the caries outcomes that demonstrated significant associations with group in the initial model. Covariables included in the adjusted models were age, sex, baseline caries experience, soft drink, lolly, syrup/jam, sugar consumption, salivary mutans streptococci and salivary LB levels. The negative binomial model was chosen due to the overdispersion of the outcome data, and the consequent inappropriateness of a Poisson model for this count data. Effect estimates are presented as Incidence Rate Ratios (IRR) with 95% confidence intervals (CIs).

Comment: Table 3 is not multivariate, but multiple. 

Response: Thank you for the suggestion. This is corrected as ‘multivariable’. 

Comment: In table 3, is the baseline caries experience predictor the total surfaces with caries? I would not think so, but then I wonder why the baseline caries surfaces from Table 1 (incipient and total) are not included in the model. 

Response: In tables 2 and 3, the variable ‘baseline caries experience’ relates to baseline caries on a specific surface and dentition conforming to the dentition and extent of caries in the outcome. For instance: when the outcome is advanced caries increment in permanent dentition, the variable baseline caries experience would only include number of surfaces with advanced caries in permanent dentition at baseline. 

Comment: Regarding table 3, if the authors are saying 2nd follow-up, then that implies they have baseline, first follow-up, and second follow-up. While I see that they included in their response to reviewer 1 that their resident statistician nixed the mixed model, you can in fact incorporate a GEE analysis for incidence or similar method to use all available data. This reviewer agrees with Reviewer 1, if you have longitudinal data, use it! 

Response: We meant the “2-year follow-up” in table 3. To account for the repeated measures nature of the data we have adjusted all regression models for caries experience at baseline. This has the effect on including both measurements of the outcome under investigation in the same model.

Comment: In the discussion, I would argue that the design of the study does not allow you to prove anything, only conclude associations. 

Response: We have reworded the Discussion to describe the findings found as associations only.

---

## [Decision Letter · Decision Letter 2]

11 Dec 2020

PONE-D-19-20492R2

Impact of a non-randomised caries preventive trial in remote Indigenous Australian children

PLOS ONE

Dear Dr. Lalloo,

Please first accept our apologies for the delay in reaching this decision. After another round of review, all positive, by three experts, I am now issuing a minor revision decision as one remaining concern, brought up during review, was not fully addressed. I expect that this should be revised easily and that acceptance should occur quickly upon resubmission without further complication or need for additional review. 

In particular, please note the concern raised by reviewer #2 in the review below, asking you to revise the language in the first paragraph of the discussion section. Namely, reviewer #2 asks that you do not make a claim of proving a hypothesis, but rather, of having evidence supportive of your hypothesis.

Relatedly, as reviewer #2 pointed out in an earlier version of this work, "impact" may be interpreted causally, which would not be appropriate given the study design and when reporting an association only. "Impact" also appears in your title and for the same reason can be misleading, as pointed out in this earlier review. I do note that you revised your manuscript to address this earlier comment, but claims of proof and impact remain in this latest version of your manuscript.

The data presented in PLOS ONE manuscripts must support the conclusions drawn (http://journals.plos.org/plosone/s/criteria-for-publication#loc-4). For this reason, please revise the corresponding passages and title to make clear that you are only reporting associations, and that your evidence is supportive of your hypothesis, rather than proof of it. 

We look forward to receiving your revised manuscript.

Sincerely,

Yann Benetreau, PhD

Senior Editor, PLOS ONE

Journal Requirements:

Additional Editor Comments (if provided):

Reviewers' comments:

Reviewer's Responses to Questions

**Comments to the Author**

1. If the authors have adequately addressed your comments raised in a previous round of review and you feel that this manuscript is now acceptable for publication, you may indicate that here to bypass the “Comments to the Author” section, enter your conflict of interest statement in the “Confidential to Editor” section, and submit your "Accept" recommendation.

Reviewer #1: All comments have been addressed

Reviewer #2: All comments have been addressed

Reviewer #3: All comments have been addressed

2. Is the manuscript technically sound, and do the data support the conclusions?

Reviewer #1: (No Response)

Reviewer #2: Yes

Reviewer #3: Yes

3. Has the statistical analysis been performed appropriately and rigorously? 

Reviewer #1: (No Response)

Reviewer #2: Yes

Reviewer #3: Yes

4. Have the authors made all data underlying the findings in their manuscript fully available?

Reviewer #1: (No Response)

Reviewer #2: Yes

Reviewer #3: No

5. Is the manuscript presented in an intelligible fashion and written in standard English?

Reviewer #1: (No Response)

Reviewer #2: Yes

Reviewer #3: Yes

6. Review Comments to the Author

Reviewer #1: (No Response)

Reviewer #2: Largely good to go

I think I'd just suggest the authors revise their language in Paragraph 1 of the discussion about "proving" their hypothesis. That hypothesis seems to be the standard alternative hypothesis. While I suppose one could argue the language isn't technically wrong, it is certainly...unusual. Proving something is difficult even in the best of circumstances, to say nothing of a nonrandomized single study. Your evidence is SUPPORTIVE of the hypothesis.

Reviewer #3: The manuscript brings essential information about a specific population. The manuscript is sound and relevant. I think this is now ready to be published.

7. PLOS authors have the option to publish the peer review history of their article (what does this mean?). If published, this will include your full peer review and any attached files.

Reviewer #1: No

Reviewer #2: **Yes: **Ryan Richard Ruff

Reviewer #3: No

---

## [Author Response · Author response to Decision Letter 2]

13 Dec 2020

PONE-D-19-20492R2

Effect of a non-randomised caries preventive trial in remote Indigenous Australian children. 

Yann Benetrau, Senior Editor, PLOS ONE

Dear Yann;

We have edited the manuscript as per the recommendations of yourself and reviewer 2. We have replaced ‘impact’ with effect or effectiveness where appropriate, including in the title.

We have revised paragraph 1 of the Discussion.

Editor comments:

In particular, please note the concern raised by reviewer #2 in the review below, asking you to revise the language in the first paragraph of the discussion section. Namely, reviewer #2 asks that you do not make a claim of proving a hypothesis, but rather, of having evidence supportive of your hypothesis.

Relatedly, as reviewer #2 pointed out in an earlier version of this work, "impact" may be interpreted causally, which would not be appropriate given the study design and when reporting an association only. "Impact" also appears in your title and for the same reason can be misleading, as pointed out in this earlier review. I do note that you revised your manuscript to address this earlier comment, but claims of proof and impact remain in this latest version of your manuscript.

Reviewer 2 comment:

I think I'd just suggest the authors revise their language in Paragraph 1 of the discussion about "proving" their hypothesis. That hypothesis seems to be the standard alternative hypothesis. While I suppose one could argue the language isn't technically wrong, it is certainly...unusual. Proving something is difficult even in the best of circumstances, to say nothing of a nonrandomized single study. Your evidence is SUPPORTIVE of the hypothesis.

We hope this is acceptable.

Best wishes and regards, 

A/ Prof Ratilal Lalloo

Corresponding author

---

## [Editor Report · Decision Letter 3]

20 Dec 2020

Effect of a non-randomised caries preventive trial in remote Indigenous Australian children

PONE-D-19-20492R3

Dear Dr. Lalloo,

We’re pleased to inform you that your manuscript has been judged scientifically suitable for publication and will be formally accepted for publication once it meets all outstanding technical requirements. 

Thank you for revising your title and discussion to address the comments previously raised in a review due to the non-randomized study design. Please note that after further consideration, although we are now issuing a provisional acceptance for publication, we are asking that you edit your manuscript as follows before submitting the final version so as to further address the concerns raised during peer-review:

in order to avoid a causal interpretation of your findings, we request that you do not refer to "effect" in your title. We suggest instead the following title: "Carious lesions in permanent dentitions are reduced in remote Indigenous Australian children taking part in a non-randomised preventive trial."for the same reasons, we request that you add in the manuscript's Discussion a section discussing the potential impact on the results of the lack of prospective randomized recruitment; this is particularly important since there are pre-existing differences between the two groups (according to Table 1). We did notice that you addressed this limitation in your Results section, but we request that this be further covered in the Discussion as well. 

Thank you for addressing the above requests in the final version of your manuscript.

Sincerely,

Yann Benetreau, PhD

Senior Editor, PLOS ONE
---

## [Editor Report · Acceptance letter]

13 Jan 2021

PONE-D-19-20492R3 

Carious lesions in permanent dentitions are reduced in remote Indigenous Australian children taking part in a non-randomised preventive trial 

Dear Dr. Lalloo:

I'm pleased to inform you that your manuscript has been deemed suitable for publication in PLOS ONE. Congratulations! Your manuscript is now with our production department. 

Kind regards, 

on behalf of

Dr. Yann Benetreau 

Staff Editor

PLOS ONE